# Can Performant LLMs Be Ethical?
# Quantifying the Impact of Web Crawling Opt-Outs

**Dongyang Fan**$^\heartsuit$**, Vinko Sabolčec**$^\heartsuit$**, Matin Ansaripour**$^\heartsuit$**, Ayush Kumar Tarun**$^\heartsuit$
**Martin Jaggi**$^{\heartsuit\dagger}$**, Antoine Bosselut**$^{\heartsuit\dagger}$**, Imanol Schlag**$^{\diamondsuit\dagger}$
$^\heartsuit$EPFL, Switzerland, $^\diamondsuit$ETH Zürich, Switzerland
`firstname.lastname@epfl.ch`, `ischlag@ethz.ch`

## Abstract

The increasing adoption of web crawling opt-outs by copyright holders of online content raises critical questions about the impact of data compliance on large language model (LLM) performance. However, little is known about how these restrictions (and the resultant filtering of pretraining datasets) affect the capabilities of models trained using these corpora. In this work, we conceptualize this effect as the *data compliance gap* (DCG), which quantifies the performance difference between models trained on datasets that comply with web crawling opt-outs, and those that do not. We measure the data compliance gap in two settings: pretraining models from scratch and continual pretraining from existing compliant models (simulating a setting where copyrighted data could be integrated later in pretraining). Our experiments with 1.5B models show that, as of January 2025, compliance with web data opt-outs does not degrade general knowledge acquisition (close to 0% DCG). However, in specialized domains such as biomedical research, excluding major publishers leads to performance declines. These findings suggest that while general-purpose LLMs can be trained to perform equally well using fully open data, performance in specialized domains may benefit from access to high-quality copyrighted sources later in training. Our study provides empirical insights into the long-debated trade-off between data compliance and downstream model performance, informing future discussions on AI training practices and policy decisions. Our website is available at `https://data-compliance.github.io/`.

## 1 Introduction

The success of Large Language Models (LLMs) is largely dependent on web-scale data. As data becomes an increasingly valuable business asset, companies are treating it as part of their Intellectual Property (IP). However, whether training LLMs on copyrighted data qualifies as fair use remains an open question. In a statement from Library Copyright Alliance (LCA): the ingestion of copyrighted works to create LLMs or other AI training databases generally is a fair use; however, if a work created by AI is substantially similar in protected expression to a work that was ingested by the AI, then it is considered copyright infringement (LCA, 2023).

One of the most notable cases in this debate is the lawsuit filed by The New York Times against OpenAI and Microsoft[1], alleging the unauthorized use of its articles to train GPT models. The accusation is two-fold: first, ChatGPT can generate verbatim excerpts from its copyrighted articles; second, ChatGPT has also hallucinated articles attributed to the Times. This case brings copyright infringement and AI development to the forefront. Amid ongoing legal and ethical debates, AI companies offer an opt-out mechanism for content publishers. Publishers can modify their Robots Exclusion Protocol (REP), commonly known as robots.txt, to block web crawlers at their discretion.

---

† denotes equal supervision
[1]nytco-assets.nytimes.com/2023/12/Lawsuit-Document-dkt-1-68-Ex-J.pdf

Starting from mid 2023, there is a rapidly increasing trend for content owners to apply restrictions on web crawling, as shown in Figure 1. This results in 5-7% token loss among the major web crawl corpora – C4 (Raffel et al., 2020), RefinedWeb (Penedo et al., 2023), and Dolma (Soldaini et al., 2024) – in April 2024. We refer readers to the work from Longpre et al. (2024) for a more detailed analysis.

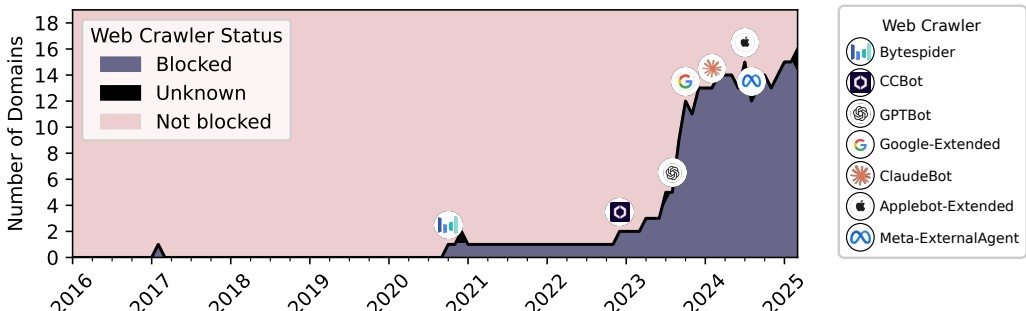

Figure 1: Timeline of the number of Top 20 filtered (based on robots.txt compliance) domains (excluding `theguardian.com`[2]) that block at least one of the web crawlers listed in Appendix B. Robots.txt files were retrieved via the Internet Archive API for each domain on a monthly basis from January 2016 to March 2025. Images indicate the first instance in which a crawler is *explicitly* blocked by any of the domains.

Ensuring legal compliance across the full LLM pipeline—from data preparation to training and inference—begins most effectively at the data preparation stage, by excluding content from websites that have opted out of web crawling. Although largely believed high-quality sources such as news articles are often found in non-permissible data, the impact of excluding major publishers on LLM performance remains poorly understood. This raises a critical question: *Can we quantify the effect of content publisher opt-outs? And if so, can we identify the specific knowledge gaps introduced by adhering to robots.txt restrictions?*

In this work, we make a first step towards answering this question. We show that, as of January 2025, content publishers opting out does not alter the training corpus data distribution much. This results in minimal performance gaps observed by complying with crawling opt-outs. Our contributions are as follows:

- We provide a thorough inspection of the change in FineWeb-Edu corpus by respecting web crawling opt-out (Section 3).
- We introduce the concept of Data Compliance Gap (DCG), which quantifies the performance gap between models trained with and without respect for web crawling opt-outs and propose two methods for measuring it (Section 4).
- We demonstrate that DCG is close to 0% for general knowledge acquisition, however, compliance gap exists in knowledge of veracity and in structural formats (Section 5.2). A noticeable DCG also exists for non-compliant medical domain data (Section 5.3).
- We show that pretraining in compliance with robots.txt reduces memorization of copyrighted content, though it also limits the acquisition of knowledge derived from that content (Section 5.4).

## 2 Related Works

**AI and Fair Use.** Fair use allows users to utilize copyrighted material without explicit permission under specific conditions. To prevent copyright infringement—especially given the strong memorization abilities of large language models (LLMs)—a variety of methods

---

[2]`theguardian.com` was not accessible via the Internet Archive API

have been proposed to support fair use across different stages of model development. Prior to model training, data filtering is the most straightforward strategy to limit the use of non-permissible content. For instance, AlphaCode (Li et al., 2022) excluded GitHub code based on license types, while Apple Foundation Models (Gunter et al., 2024) honor websites' opt-out requests using standard robots.txt directives to prevent crawling by Applebot. During model training, techniques like differentially private training (Carlini et al., 2021; Mattern et al., 2022) help prevent the extraction of personal data. Source-aware training (Khalifa et al., 2024) associates source information with each document, enabling instance-level attribution. More recently, approaches like training with Goldfish loss (Hans et al., 2024) have been introduced to effectively reduce verbatim memorization. For models that have already been trained on copyrighted content, post-training techniques such as Reinforcement Learning from Human Feedback (RLHF) and output filtering (Ippolito et al., 2023) have been studied. Another area that is gaining traction for mitigating the generation of privacy-sensitive or copyrighted material in post-training is model unlearning (Golatkar et al., 2020; Chundawat et al., 2023b;a; Tarun et al., 2023a;b; Shi et al., 2025). Our work falls into the first category, as we respect robots.txt compliance from the data preparation stage.

**Data Valuation.** In the Machine Learning field, data valuation is to understand the impact of data in downstream tasks. Koh & Liang (2020) popularized the Leave-One-Out (LOO) valuation (Weisberg & Cook, 1982) from statistics to measure data impact for black-box ML approaches. Inspired by cooperative game theory, Data Shapley (Ghorbani & Zou, 2019) is another popular approach for measuring data importance, which offers a more theoretically fair attribution of data value compared to LOO. However, it is significantly more computationally intensive than LOO, as it requires retraining over all possible data subsets. While LOO offers a more tractable alternative, it still involves model retraining, making it costly for large-scale models (Fan et al., 2025). In the LLM world, more computationally feasible methods such as influence functions (Kwon et al., 2024; Choe et al., 2024; Xia et al., 2024) are more often used. Yet these methods often rely on gradient alignment and model loss is not directly indicative for downstream performance for LLMs. Thus, these methods can be inaccurate. In this work, aiming for a more accurate measurement, we adopt LOO for evaluating the impact of robots.txt excluded data, and we propose more computationally feasible alternatives specifically designed for LLM pretraining.

# 3   Data Inspection

We analyze the FineWeb-Edu dataset (Lozhkov et al., 2024), excluding any documents that, as of January 2025, disallow crawling to the bots listed in Appendix B in their robots.txt file. While FineWeb-Edu is derived from CommonCrawl[3]—which dates back to 2008 and respects robots.txt restrictions—those restrictions are enforced only at the time of crawling. Since website owners can modify their robots.txt files at any time, CommonCrawl may include content from sites that currently block crawlers but did not at the time of data collection. To ensure full compliance, we apply **retrospective filtering** using the most recent robots.txt files, resulting in a dataset that adheres to current crawling restrictions. We refer to the original, unfiltered dataset as *non-compliant*, and the filtered version as *compliant*. In comparison to the non-compliant corpus, the compliant version contains 8% fewer training tokens. This ratio is consistent with what was reported in Longpre et al. (2024).

**Top Filtered Domains.**   Comparing the non-compliant and compliant data corpora, we present a list of the Top 20 filtered URL domains in Figure 2, sorted by document count differences between the two corpora. We used ChatGPT to help get the taxonomy and refined it via human inspection. The top URL domains are largely centered around the following topics: *News & Media*, *Science & Technology*, *Health & Medical Information*.

**Opt-out Timelines for the Top Domains.**   We collected historical snapshots of the robots.txt files for the Top 20 domains and plotted the first recorded instance of each domain blocking any of the AI training crawling bots (as listed in Appendix B) in Figure 1. Notably, Bytespider

---

[3]commoncrawl.org

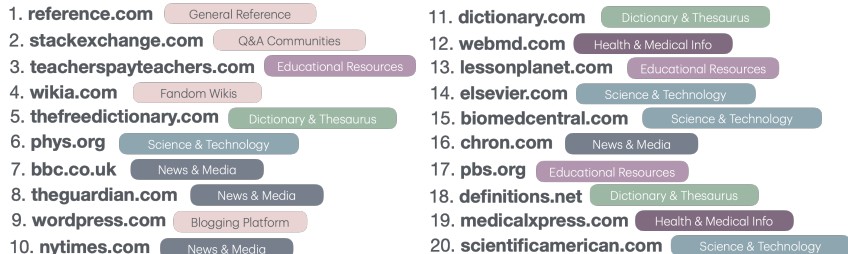

Figure 2: Top 20 filtered URL domains based on differences in document counts between the compliant and non-compliant corpora, along with their corresponding taxonomy.

bot by ByteDance was already being blocked as early as 2021. Since mid-2023, the OpenAI bot began to appear in blocklists. By mid 2024, more major tech companies had started facing blocks across these top domains.

**Data Distribution Change After Robots.txt Filtering.** To assess whether adhering to the opt-out process affects the data distribution, we used WebOrganizer (Wettig et al., 2025) to cluster documents based on both topic and format, two orthogonal features representing the subject of content (topic) and the form of the content (format). We then visualized the distribution of these clusters in Figure 3. Interestingly, the overall distribution differences between compliant and non-compliant data are minimal, with the notable exception of a decrease in the proportion of *News Articles* and *Science & Tech.* and an increase in *Personal Blogs*. For completeness, the percentage decrease in each category is presented in Appendix C.

**Reduced Occurrence of Copyrighted Data.** To further validate our data processing pipeline, we searched for matching 50-grams of data from `nytimes.com`, `medicalxpress.com`, and `stackexchange.com` domains between the compliant and non-compliant corpora. While the frequency of copyrighted articles is reduced in the compliant corpus, it is not entirely eliminated. Our manual inspection of the compliant corpus found that many remaining overlaps in `nytimes.com` data result from republished content, while matched `medicalxpress.com` content in the compliant dataset is itself republished from other sources; in contrast, `stackexchange.com` content primarily consists of references to other websites and is not republished. We provide examples of matched 50-grams found in the compliant dataset in Appendix D.

Table 1: Percentage of documents that have a 50-gram match with documents coming from `nytimes.com`, `medicalxpress.com`, or `stackexchange.com` domains in compliant and non-compliant FineWeb-Edu training corpora

|  | nytimes.com | medicalxpress.com | stackexchange.com |
|---|---|---|---|
| Compliant corpus | 0.267% | 0.637% | 0.348% |
| Non-compliant corpus | 0.354% | 0.702% | 0.539% |

## 4 Data Compliance Gap

While we have observed differences between the compliant and non-compliant training corpora, it remains unclear how these differences affect the performance of a language model. In this section, we introduce the concept of the **Data Compliance Gap (DCG)** to characterize the impact of data compliance on model performance.

### 4.1 Definition

We define DCG as the downstream performance gap between the compliant and non-compliant trained models. The difference between compliant and non-compliant trained

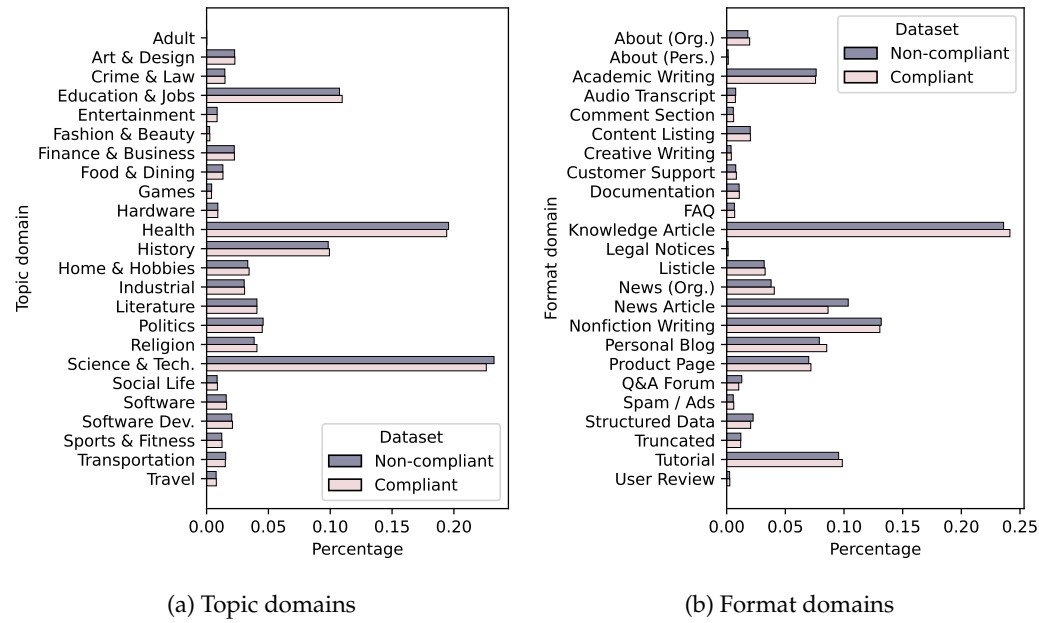

(a) Topic domains  (b) Format domains

Figure 3: Distribution of WebOrganizer (Wettig et al., 2025) topic and format domains annotated in non-compliant and compliant FineWeb-Edu corpora.

models should *only* be the training corpus, i.e. the compliant version is derived by filtering the non-compliant corpus according to robots.txt restrictions.

DCG depends on the downstream task domain and the training strategy, including model architecture, size, and the volume of training data.

### 4.2 Settings

We measure DCG using the standard Leave-One-Out approach: excluding non-compliant data completely from training to assess DCG under robots.txt restrictions (M1), or reintegrating non-compliant data into compliant training to approximate the DCG (M2). The two settings are presented in Figure 4.

**M1.** Conduct pre-training runs using compliant and non-compliant corpora, respectively. The resulting differences in downstream performance define the DCG. M1 is an exact measure of DCG, yet can be computationally expensive.

**M2.** From a pre-trained compliant checkpoint, conduct continual pre-training with an annealing learning rate (cooldown) using compliant and non-compliant data. The resulting downstream performance differences are thus DCG. M2 is an approximate measure, which is more computationally feasible.

## 5 Experiments

### 5.1 Experimental Setup

**Model.** We adopt the Llama model architecture (Grattafiori et al., 2024) with 16 layers, a hidden size of 2048, a sequence length of 4096, and a batch size of 2.06 million, totaling 1.5 billion parameters. To enable cooldown experiments without retraining models from scratch, we follow the WSD learning schedule (Hu et al., 2024), applying 2000 warmup steps and 4844 cooldown steps. We use the AdamW optimizer with regularization strength 0.1 (Loshchilov & Hutter, 2019) with a learning rate of 3e-4 and cool down to learning rate 3e-5. To train the models, we use the Megatron-LM framework (Narayanan et al., 2021).

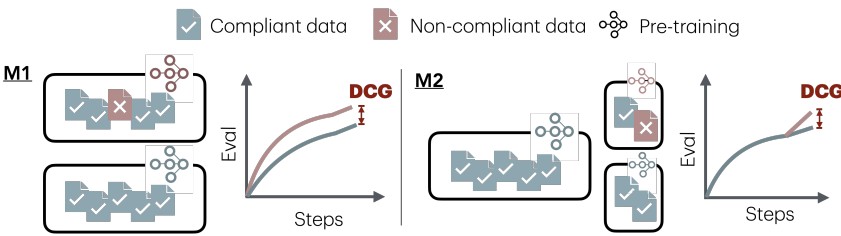

Figure 4: Different settings of estimating DCG. For both settings, the difference between compliant and non-compliant trained models only exists in the data, i.e. if non-compliant data are involved in model training. DCG measures the downstream eval differences between the trained models.

## 5.2 Compliance Gap Measured from Training from Scratch

We pre-train using the compliant and non-compliant corpora respectively. To evaluate DCG, we focus on the following evaluation benchmarks.

**Benchmarks for General Knowledge**. As standard practice, we evaluate models on general knowledge understanding using LM-Eval-Harness developed by Gao et al. (2024). The benchmarks used are Arc-Easy (Clark et al., 2018), Arc-Challenge (Clark et al., 2018), CommonSense QA (CSQA, Talmor et al., 2018), OpenBook QA (OBQA, Mihaylov et al., 2018), MMLU (Hendrycks et al., 2020), PIQA (Bisk et al., 2020), Social IQA (SIQA, Sap et al., 2019), HellaSwag (HS, Zellers et al., 2019), Lambada (LBD, Paperno et al., 2016) and Winogrande (WG, Sakaguchi et al., 2021).

**Benchmarks for Different Knowledge Categories.** We use the Pinocchio dataset from Hu et al. (2023), which covers factual knowledge of the following aspects. Note these aspects are not disjoint, for example, a temporal fact can be stored in a structural format as well.

- *Temporal.* Questions based on modifications made to factual content in Wikipedia articles. The goal is to test if LLMs are capable of discerning factual knowledge from different time periods.
- *Structural.* Questions are based on Wikipedia articles. The goal is to test if LLMs can memorize and reason over facts from structured formats (tables, lists, or databases).
- *Multifaceted.* Questions are based on Wikipedia articles. The goal is to test if LLMs can memorize and reason over multiple pieces of information obtained during pre-training.
- *Adversarial.* Questions curated from Symmetric (Schuster et al., 2019) and FM2 (Eisenschlos et al., 2021) that are strategically modified to deceive advanced LLMs. The goal is to examine whether LLMs can withstand adversarial examples in the context of factuality.
- *Domain Specific.* The dataset covers samples from PubHealth (Kotonya & Toni, 2020) in the public health domain and SciFact (Wadden et al., 2022) in the science domain. The goal is to test if LLMs can reason over domain-specific factual knowledge.
- *Real World.* Questions based on Politifact (Misra, 2022), focusing primarily on political claims. The goal is to test if LLMs understand real-world political facts.

Table 2: Compliance gap evaluated over standard common knowledge benchmarks between 1.5B models trained from scratch on 100B tokens. Each row denotes a different model trained from scratch.

|  | Arc-C | Arc-E | CSQA | OBQA | MMLU | PIQA | SIQA | HS | LBD | WG | Avg |
|---|---|---|---|---|---|---|---|---|---|---|---|
| Non-compliant | 34.1 | 70.0 | 20.8 | 27.4 | 32.0 | 71.5 | 40.4 | 42.0 | 34.7 | 52.2 | 42.5 |
| Compliant | 32.8 | 69.1 | 20.2 | 26.0 | 32.0 | 71.0 | 41.5 | 42.0 | 35.4 | 57.5 | 42.8 |
| -News | 35.1 | 70.1 | 19.8 | 26.6 | 31.8 | 71.8 | 40.4 | 42.4 | 36.0 | 56.0 | 43.0 |

Table 2 shows no significant difference in general knowledge performance between compliant and non-compliant pre-trained models. However, Figure 5 reveals a noticeable compliance gap in structural and adversarial knowledge. This likely stems from the fact

that structural formats are prevalent in documents from Math & Science and Biomedical domains, and resistance to falsified facts can be enhanced through the inclusion of newspaper articles and scientific papers—areas often associated with organizations that restrict web crawling.

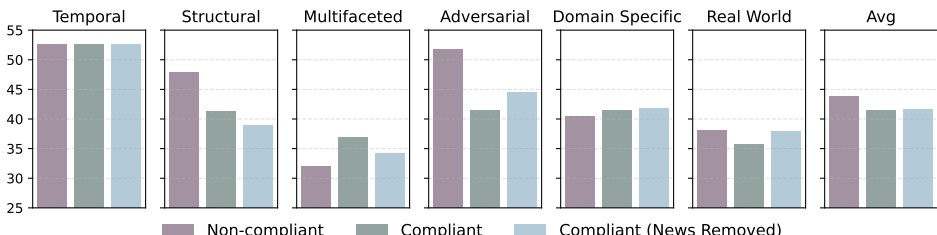

Figure 5: Compliance gap within different sources of factual knowledge from Pinocchio Benchmark (Hu et al., 2023) after pre-training on 100B tokens. The compliance gap is more prominent in structural (from tables, lists, or databases) and adversarial knowledge.

**What If All News Publishers Opt Out?**   We have previously shown that, as of January 2025, adhering to robots.txt does not lead to a significant change in performance. We hypothesize that a large number of smaller publishers help compensate for the knowledge lost from major publishers who have opted out. But what happens if this trend continues and smaller publishers also begin to opt out? We aim to investigate whether this would have a measurable impact.

Since obtaining a comprehensive list of all URLs within a specific domain is challenging, we rely on existing sources. Currently, a critical domain is newspapers. To simulate a scenario where all newspaper publishers opt out, we use a list of 1,158 news publishers curated by `homepages.news`[4], which serves as a reasonable proxy for the broader set of news domains. We then exclude all newspaper articles from these domains in our *compliant* corpus, resulting in an approximate 4% reduction in total tokens. As shown in Appendix D, the republication of news articles is quite common. To accurately assess their impact, we further perform a decontamination step that removes document segments replicated from news articles hosted on different URL domains. This process results in a further 14% reduction in total compliant tokens. After this process, the proportion of documents containing a 50-gram match with content from `nytimes.com` drops to 0.012%, indicating the effectiveness of the decontamination.

Interestingly, even after excluding all newspaper articles, we observe almost no change in downstream performance, as shown in Table 2. Moreover, in the Pinocchio-Temporal and Pinocchio-Real World tasks—where newspaper articles would be expected to perform well—we observe no drop in performance even after removing all newspaper content. We believe this is due to factual knowledge being present from other sources.

> **Takeaway 1.** Pre-training on fully open data does not significantly impact general knowledge understanding. Even if all news publishers opt out, the effect remains minimal.

> **Takeaway 2.** Compared to training on non-compliant data, compliant pretraining tends to reduce a model's ability to understand structural knowledge (i.e., information presented in structured formats) and its robustness against falsified knowledge (strategically crafted to mislead advanced LLMs).

---

[4]https://palewi.re/docs/news-homepages/openai-gptbot-robotstxt.html

> **Takeaway 3.** Newspaper articles may not be as important as expected. Temporal and real-world knowledge, which are prominently featured in newspaper articles, can be supplemented through other sources.

### 5.3 Compliance Gap Measured from Continual Pre-Training

Continual pretraining has been the standard paradigm for models to efficiently adapt to emergent new knowledge or a different target distribution (Parmar et al., 2024; Çağatay Yıldız et al., 2025). In this section, we simulate the scenario where copyrighted data can be integrated later into pretraining. We take the *compliant* 1.5B model checkpoint after training on 90B/1.6T tokens and continually pre-train with 10B/100B tokens. We use an annealing learning rate with a linear cooldown schedule for the continual pretraining phase.

**Major Non-compliant Domains.** Based on the taxonomy in Section 3, we focus on the major domains identified among the Top 50 filtered URLs: News, Medical, and Math & Science. We categorize the Top 50 URL domains into the aforementioned three groups, and construct three domain-specific sets of non-compliant data, detailed in Appendix E.1. To perform a fine-grained ablation, we compare the performance of compliant continual pretraining with that of continual pretraining on the same compliant dataset (4.5T tokens) augmented by reintegrating domain-specific non-compliant documents. The proportion of non-compliant tokens added reflects their natural distribution in the original corpus.

**Domain-specific Benchmarks.** The aim of this evaluation is to test whether the inclusion of certain non-compliant data later in training can enhance domain-specific knowledge acquisition. Towards this goal, we selected the following benchmarks: 1) **Concurrent and temporal factual knowledge**. We select *Reuters-QA* (Muhlgay et al., 2023), which is based on Reuters articles published after 1/10/2021, and the *Temporal QA* category from Pinocchio benchmark. 2) **Medical knowledge**. We select *PubMedQA* (Jin et al., 2019), which contains biomedical knowledge curated from biomedical research publications, and *Domain-Specific QA* from the Pinocchio benchmark, which covers samples from PubHealth. 3) **Scientific knowledge**. We select SciQ (Welbl et al., 2017), which is curated from 28 books covering biology, chemistry, earth science, and physics and spanning elementary level to college introductory material.

Table 3 presents the domain-specific cooldown results. The underlined scores correspond to runs where task-specific, non-compliant data was added during cooldown, which is supposed to enhance performance on that specific task. The remaining entries illustrate expected variability across cooldown runs without such targeted augmentation. Among the three domains, only the Medical tasks with long training (1.7T tokens) show a consistent signal. A similar pattern emerges with the Pinocchio benchmarks, where reintroducing non-compliant Medical data leads to the most pronounced compliance gap, as shown in Figure 6.

Table 3: Domain-specific evaluation results are shown across various cooldown checkpoints. Underlining indicates instances where the non-compliant domain-specific data aligns with the knowledge assessed in the target domain.

|  | Reuters-QA | Pnc-Temporal-QA | PubMedQA | Pnc-DomainSpecific-QA | SciQ |
|---|---|---|---|---|---|
| *100B Tokens Trained* |  |  |  |  |  |
| Compliant | 52.9 | 53.0 | 57.4 | 41.5 | 88.4 |
| Compliant + News | 52.5 | 54.2 | 57.2 | 41.3 | 89.4 |
| Compliant + Med | 51.7 | 54.3 | 57.0 | 42.1 | 88.8 |
| Compliant + MathSci | 52.7 | 52.8 | 56.6 | 41.6 | 89.0 |
| *1.7T Tokens Trained* |  |  |  |  |  |
| Compliant | 65.1 | 52.8 | 61.4 | 44.5 | 90.0 |
| Compliant + News | 64.6 | 54.1 | 61.2 | 44.7 | 90.2 |
| Compliant + Med | 65.1 | 53.0 | 63.0 | 46.9 | 90.0 |
| Compliant + MathSci | 65.3 | 51.9 | 61.6 | 45.6 | 90.0 |

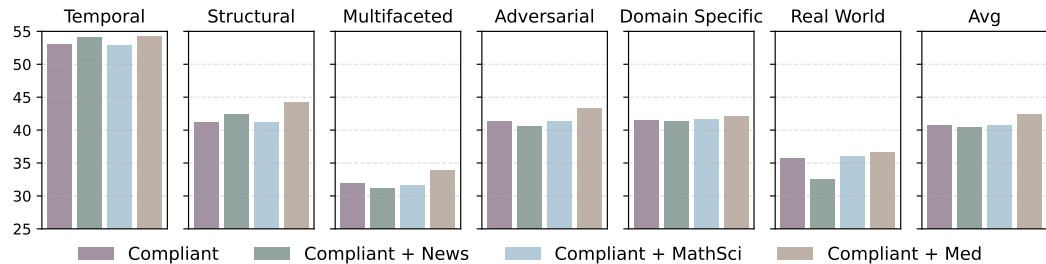

Figure 6: Compliance gap measured using the Pinocchio benchmark across various cooldown checkpoints (100B). Non-compliant medical data introduces a noticeable compliance gap.

> **Takeaway 4.** A noticeable compliance gap exists when non-compliant medical domain data is introduced at a later stage of training.

## 5.4 Reduced Memorization of Copyrighted Articles?

As data without crawling consent is removed from the corpus, we would expect compliant models to have less memorization of copyrighted content. To test this, we filter out all New York Times articles from the non-compliant corpus and measure the memorization of these articles across our trained models. We sampled 12'800 New York Times articles and for each article, we extracted the first 128, 256, and 512 tokens as prompts, then compared the models' generations against the original text. Following the setup of Freeman et al. (2024), we also measure the longest common continuous substrings (LCCS), as an indication of verbatim memorization. On top of that, we add the BLEU metric, which is calculated based on N-gram overlap.

We further curated two benchmarks from these 12'800 articles, each containing 1000 questions: (1) NYTimes Multiple Choice Completion (NYTimes-MCC) task, to test if a model can select the correct completion among multiple choices given a partial excerpt from a news article. (2) NYTimes Multiple Choice Question (NYTimes-MCQ) task, which asks factual questions derived from these articles, aiming to provide challenging questions grounded in unique event details or context found in the news coverage. The benchmarks are verified using Proprietary LLMs. We offer more details regarding the scripts used for generating the QAs and more experimental results in Appendix F.

Table 4 presents the results of the memorization assessment. For comparison, we include models from the Qwen2.5 (Yang et al., 2024) and Llama3.2 (Grattafiori et al., 2024) families, noting that larger models tend to exhibit higher levels of memorization. The compliant model demonstrates reduced memorization of copyrighted content compared to the non-compliant model, as shown by lower LCCS, BLEU, and NYT-MCC scores—a trend consistent with the removal of newspaper articles. Importantly, compliance does not necessarily lead to a loss of specific knowledge from New York Times articles, as indicated by comparable NYT-MCQ scores. Notably, even with news decontamination applied, the NYT-MCQ score does not decrease.

> **Takeaway 5.** Compliant training reduces verbatim memorization of copyrighted content without necessarily compromising specific knowledge derived from the publications.

Table 4: Memorization and factual knowledge assessment of New York Times articles across various models using 256-token prefix prompts. PT, CD, and Compl. stand for "Pretraining", "Cooldown", and "Compliant" respectively. Values in parentheses indicate the number of tokens seen in training.

| | LCCS ↓ | BLEU ↓ | NYTimes-MCC ↓ | NYTimes-MCQ ↑ |
|---|---|---|---|---|
| Qwen2.5-0.5B | 19.51 | 0.71 | 42.8 | 24.4 |
| Qwen2.5-1.5B | 24.35 | 1.18 | 52.1 | 30.7 |
| Qwen2.5-3B | 27.13 | 1.54 | 56.4 | 31.3 |
| Qwen2.5-7B | 30.99 | 1.95 | 63.8 | 36.4 |
| Llama-3.2-1.2B | 20.71 | 0.58 | 49.9 | 24.8 |
| Llama-3.2-3B | 22.75 | 0.70 | 57.6 | 30.1 |
| PT + CD (Non-compl. 100B) | 23.11 | 0.63 | 50.6 | 28.9 |
| PT + CD (Compl., 100B) | 21.27 | 0.51 | 48.6 | 27.5 |
| PT + CD (Compl., with News Removed, 100B) | 20.28 | 0.53 | 48.0 | 28.6 |
| PT (Compl., 90B) + CD (Non-Compl., 10B) | 21.74 | 0.55 | 50.4 | 26.9 |
| PT (Compl.,90B) + CD (Compl. + Non-Compl. News, 10B) | 21.33 | 0.51 | 48.4 | 28.0 |

## 6  Conclusions and Future Work

In this work, we take an initial step toward examining the widely discussed trade-off between data compliance and downstream model performance. We introduce the concept of the Data Compliance Gap (DCG) to quantify performance differences resulting from adhering to robots.txt opt-outs, and evaluate DCG across various training settings. Our findings indicate that it is currently feasible to train performant LLMs exclusively on data that respects web crawling restrictions. Thus, we recommend adherence to robots.txt restrictions for LLM developments.

We clarify that our notion of compliance addresses only part of data usage rights. Our approach focuses on AI-specific user agents, offering a clearer signal of consent than general crawling permissions. However, it does not cover all legal constraints, such as Terms of Service in "browsewrap agreements", which are legally ambiguous and challenging to process at scale. Therefore, our robots.txt filtering represents a practical upper bound on demonstrable compliance, not full legal clearance. We call for future work for a finer-grained study.

Our study is constrained by the model size and training budget feasible within an academic setting. Whether the DCG scales similarly with larger models remains an open question, which we further discuss in Appendix A. While current-day robots.txt restrictions do not lead to a significant DCG, it is yet to be seen whether the gap will widen as more content owners opt out of web crawling.

## Ethics Statement

The goal of our work is to promote ethical development of LLMs, via evaluating the impact of using non-permissible data.

## Acknowledgement

DF would like to thank Maksym Andriushchenko for his feedback on an initial draft of the paper. DF acknowledges funding from SNSF Grant number 10005248. AB gratefully acknowledges the support of the Swiss National Science Foundation (No. 215390), Innosuisse (PFFS-21-29), the EPFL Center for Imaging, Sony Group Corporation, and a Meta LLM Evaluation Research Grant. This work was supported as part of the Swiss AI Initiative by a grant from the Swiss National Supercomputing Centre (CSCS) under project ID a06 on Alps.

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

## A Limitations

Despite our efforts to construct a strictly compliant corpus, some non-permissible content may still be present. For instance, New York Times articles republished on personal blogs remain accessible, as these blogs do not block web crawlers. Similarly, websites like `medicalexpress.com`, though they restrict crawling, often rehost content from publicly accessible sources. As a result, distinguishing truly compliant from non-compliant data at the document level is challenging. Adhering to URL-specific robots.txt directives remains the most practical and enforceable approach available.

We acknowledge that robots.txt compliance addresses only one aspect of data usage rights. Our approach evaluates robots.txt rules specifically for AI training user agents, offering a clearer signal of consent than general crawling permissions. However, this does not account for all legal restrictions, such as Terms of Service in "browsewrap agreements." These terms are legally ambiguous and difficult to process at scale due to their natural language format. Thus, our robots.txt filtering reflects a practical upper bound of demonstrable compliance, not comprehensive legal clearance.

A key limitation of our study is that evaluations were conducted solely on 1B-scale models. Given their limited capacity, these models may exhibit constrained knowledge acquisition and produce noisier evaluation results. In contrast, larger models tend to memorize more readily and at a faster rate (Tirumala et al., 2022; Carlini et al., 2023; Freeman et al., 2024). At the scale used in our experiments, it is likely that the models lack sufficient capacity for effective memorization, which may explain the generally low LCCS observed in Table 4.

Moreover, due to the limited model size, we are unable to reliably evaluate domain-specific knowledge in Math & Science, which often requires reasoning capabilities. Our current

benchmark, SciQ, primarily assesses factual recall rather than reasoning. While a DCG may exist in this domain, our current setup does not allow for a rigorous evaluation.

## B List of blocked web crawlers

The list of crawler bots that we consider for data filtering.

```
# list of blocked bots
"AI2Bot",  # AI2
"Applebot-Extended",  # Apple
"Bytespider",  # Bytedance
"CCBot",  # Common Crawl
"CCBot/2.0",  # Common Crawl
"CCBot/1.0",  # Common Crawl
"ClaudeBot",  # Anthropic
"cohere-training-data-crawler",  # Cohere
"Diffbot",  # Diffbot
"Meta-ExternalAgent",  # Meta
"Google-Extended",  # Google
"GPTBot",  # OpenAI
"PanguBot",  # Huawei
"*"
```

## C Percentage decrease in each category due to data compliance

In addition to changes in data distribution, Tables 5 and 6 report the percentage of data removed by domain and relative to the entire corpus. For instance, filtering out 11.39% of Science & Tech data leads to a 2.6483% reduction in total corpus tokens.

Table 5: Topic domain removal percentages

| Topic Domain | % Removed (Domain) | % Removed (Corpus) |
|---|---|---|
| Science & Tech. | 11.39 | 2.6483 |
| Health | 9.65 | 1.8893 |
| History | 8.05 | 0.7921 |
| Education & Jobs | 7.11 | 0.7639 |
| Politics | 10.45 | 0.4778 |
| Literature | 8.94 | 0.3639 |
| Industrial | 8.02 | 0.2438 |
| Home & Hobbies | 6.01 | 0.2002 |
| Finance & Business | 8.46 | 0.1894 |
| Art & Design | 8.08 | 0.1832 |
| Transportation | 10.84 | 0.1676 |
| Religion | 3.58 | 0.1377 |
| Software Dev. | 6.52 | 0.1327 |
| Crime & Law | 8.03 | 0.1184 |
| Food & Dining | 8.46 | 0.1114 |
| Software | 6.92 | 0.1096 |
| Sports & Fitness | 7.36 | 0.0903 |
| Hardware | 9.12 | 0.0829 |
| Entertainment | 9.44 | 0.0806 |
| Social Life | 6.72 | 0.0572 |
| Travel | 6.94 | 0.0534 |
| Games | 11.69 | 0.0478 |
| Fashion & Beauty | 3.61 | 0.0089 |
| Adult | 0.81 | 0.0001 |

Table 6: Format domain removal percentages

| Format Domain | % Removed (Domain) | % Removed (Corpus) |
|---|---|---|
| News Article | 24.09 | 2.4956 |
| Knowledge Article | 6.87 | 1.6224 |
| Nonfiction Writing | 9.76 | 1.2853 |
| Academic Writing | 9.57 | 0.7303 |
| Tutorial | 5.95 | 0.5673 |
| Product Page | 6.53 | 0.4566 |
| Structured Data | 17.39 | 0.3920 |
| Q&A Forum | 27.33 | 0.3518 |
| Listicle | 6.62 | 0.2115 |
| Content Listing | 8.51 | 0.1708 |
| Personal Blog | 1.68 | 0.1330 |
| Truncated | 10.22 | 0.1236 |
| News (Org.) | 2.49 | 0.0942 |
| Audio Transcript | 12.23 | 0.0940 |
| Documentation | 7.58 | 0.0804 |
| FAQ | 5.13 | 0.0336 |
| Comment Section | 4.13 | 0.0233 |
| Customer Support | 2.88 | 0.0223 |
| About (Org.) | 1.15 | 0.0208 |
| Creative Writing | 4.09 | 0.0154 |
| Spam / Ads | 2.17 | 0.0122 |
| User Review | 4.09 | 0.0103 |
| About (Pers.) | 1.85 | 0.0023 |
| Legal Notices | 1.67 | 0.0020 |

## D Matched 50-grams in the compliant dataset

We show parts of documents that contain matched 50-grams (in bold) from the `nytimes.com`, `medicalxpress.com`, and `stackexchange.com` domains in the compliant FineWeb-Edu corpora.

### D.1 `nytimes.com`

Part of an article quoted on `unilad.co.uk` (unavailable)

[...] The billionaire writes in the The New York Times:
**Every one of us — citizens, philanthropists, business and government leaders — should be troubled by the enormous gap between how little of our natural world is currently protected and how much should be protected. It is a gap that we must urgently narrow, before our human footprint consumes the earth's** remaining wild places. [...]

Republished article on `alessandrosicurocomunication.com`

**Google and a corporation associated with NASA are forming a laboratory to study artificial intelligence by means of computers that use the unusual properties of quantum physics. Their quantum computer, which performs complex calculations thousands of times faster than existing supercomputers, is expected to be in active use in the** third quarter of this year. The Quantum Artificial Intelligence Lab, as the entity is called, will focus on machine learning, which is the way computers take note of patterns of information to improve their outputs. [...]

> **Republished article on `bike.enginerve.com`**
>
> In a post today on the NYTimes there was an interesting piece on the nature of cycling and head injuries. I still object to laws requiring a helmet to be worn at all times and wear my helmet constantly for touring and commuting. Assumption of risk, and cost, is an issue and not one to be lightly avoided. And neither are the responsibility to utilize appropriate safety gear.
> Cycling Is the Top Sport for Head Injuries
> Anahad O'Connor tackles health myths in this NY Times Post.
> **Last week, New York City began its long-awaited bicycle sharing program, the largest in the nation. As in many other cities, helmet use was made optional, in part to encourage greater participation. But a look at the statistics suggests that riding without a helmet is not a decision to** [...]

## D.2 `medicalxpress.com`

> **`medicalxpress.com` and `healthed.com.au` republished Stanford University Medical Center**
>
> **People suffering from a debilitating and often discounted disease known as chronic fatigue syndrome may soon have something they've been seeking for decades: scientific proof of their ailment. Researchers at the Stanford University School of Medicine have created a blood test that can flag the disease, which currently lacks** a standard, reliable diagnostic test. "Too often, this disease is categorized as imaginary," said Ron Davis, PhD, professor of biochemistry and of genetics. When individuals with chronic fatigue syndrome seek help from a doctor, they may undergo a series of tests that check liver, kidney and heart function, as well as blood and immune cell counts, Davis said. [...]

> **`medicalxpress.com` and `hamodia.com` (unavailable) republished The Associated Press**
>
> **Since late last year, people in the central Chinese city of Wuhan have been infected with a viral pneumonia whose cause was unknown. The outbreak raised the specter of another SARS epidemic, which killed hundreds in 2002 and 2003. A preliminary investigation has now identified the respiratory disease as a** new type of coronavirus, Chinese state media reported Thursday, citing scientists handling the investigation. As of Sunday, local authorities reported 59 people with the illness. Seven were in critical condition, while the rest were stable. Eight were discharged Wednesday night after they didn't exhibit any more symptoms for several days. [...]

## D.3 `stackexchange.com`

> **Quote found on `www-cs-faculty.stanford.edu` matches quoted answer**
>
> [...] **Many readers are no doubt thinking, "Why does Knuth replace MIX by another machine instead of just sticking to a high-level programming language? Hardly anybody uses assemblers these days." Such people are entitled to their opinions, and they need not bother reading the machine-language parts of my books.** But the reasons for machine language that I gave in the preface to Volume 1, written in the early 1960s, remain valid today: [...]

> **Poetry found on en.wikiquote.org matches quoted answer**
>
> [...]
> **And the Raven, never flitting,**
> **Still is sitting, still is sitting**
> **On the pallid bust of Pallas**
> **Just above my chamber door;**
> **And his eyes have all the seeming**
> **Of a demon's that is dreaming,**
> **And the lamplight o'er him streaming**
> **Throws his shadow on the floor,**
> **And** my soul from out that shadow,
> That lies floating on the floor,
> Shall be lifted—nevermore.
> [...]

# E   Additional experimental results

## E.1   Top 50 filtered URLs

- News Domains: `bbc.co.uk`, `theguardian.com`, `nytimes.com`, `chron.com`, `abc.net.au`, `washingtonpost.com`, `theatlantic.com`, `wired.com`, `businessinsider.com`, `forbes.com`. This in total accumulates to around 1B tokens.
- Medical Domains: `webmd.com`, `biomedcentral.com`, `medicalxpress.com`, `healthline.com`, `psychcentral.com`, `medicalnewstoday.com`, `medicinenet.com`, `emedicinehealth.com`, `jamanetwork.com`. This in total accumulates to around 10B tokens.
- Math & Science Domains: `mathhelpforum.com`, `mathworks.com`, `stackexchange.com`, `scientificamerican.com`, `elsevier.com`, `windows2universe.org`. This in total accumulates to around 1.4B tokens.

## E.2   More results

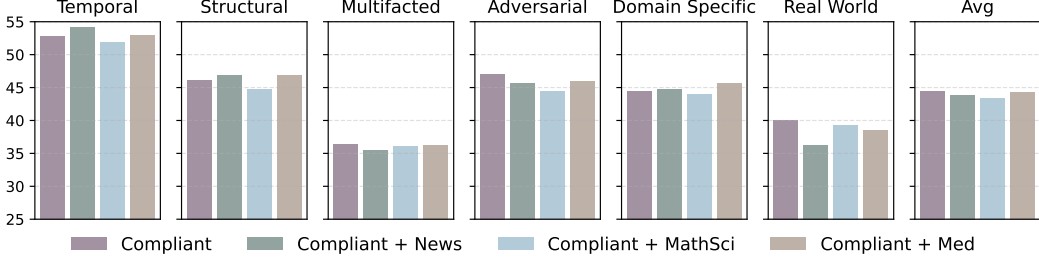

Figure 7: Compliance gap measured using the Pinocchio benchmark across various cooldown checkpoints (1.7T tokens). DCG is less prominent with longer training.

# F   NYTimes Memorization Experiments

In this section, we provide additional details about our experiments measuring the memorization of New York Times articles. The underlying datasets are not publicly released due to copyright limitations.

### F.1 Benchmark Creation Prompts

We used DeepSeek-R1 to generate the NYTimes-MCC and NYTimes-MCQ benchmarks.

- **NYTimes-MCQ Prompt:** We instructed the model to generate a self-contained multiple-choice question that requires synthesizing information across the provided context.

- **NYTimes-MCC Prompt:** We instructed the model to extract a meaningful prefix from an article and create a multiple-choice task with four possible continuations. Only one continuation (the original text from the article) is correct, while the other three are minimal edits containing specific types of factual errors.

We gathered 1000 questions for each benchmark to make the benchmarks more robust to noise. Here, you can see the prompts that we used to create the questions for each article.

---

**NYTimes-MCQ Prompt**

```
 Create a multiple-choice question that requires synthesizing information
across the provided context.  While the correct answer should only be
determinable by analyzing and combining specific details from the context,
the question itself must be self-contained and clearly understandable to
someone who hasn't read the context.

Your question should 1) test the ability to connect related information
from different parts of the context, 2) be completely clear and unambiguous
as a standalone question, 3) avoid referencing the context directly, e.g.,
no "according to the passage" phrasing.

The question should have 4 options, one of which is the correct
answer.  Please explain how to reach the correct answer from the given
context.

You are communicating with an API, not a user.  Please output in
JSON format. Begin all AI responses with the character { to produce valid
JSON. Here is an example:
{
"question": "<question>",
"A": "<option1>",
"B": "<option2>",
"C": "<option3>",
"D": "<option4>",
"answer": "<correct_option>",
"explanation": "<explanation>"
}

Here is an example:

<the example>

<the article>
```

---

---

**NYTimes-MCC Prompt**

```
 Given the provided context, create a multiple-choice context completion
task. Extract a prefix from the given context and provide four possible
completions, where only one is factually correct.  While the correct
completion should only be determinable by analyzing the provided context,
the prefix itself must be self-contained with information and clearly
understandable to someone who hasn't read the context.

The prefix must be an exact excerpt from the given context.  The
correct completion must be the original continuation of this prefix in the
given context. The three incorrect completions should be minimal edits of
the correct completion, each containing a contradiction and specific type
of factual error while remaining grammatical and fluent.

The incorrect completions should incorporate these error types:
1) Predicate error: Modify a verb or action that makes the completion
factually inconsistent.
2) Entity error: Replace a subject or object with an incorrect entity that
creates a factual inconsistency.
3) Circumstance error: Change information about location, time, or manner
that introduces a factual error.
4) Coreference error: Modify a pronoun or reference to point to a wrong or
non-existing entity.
5) Link error: Change how statements are linked together (causal/temporal
links) to create a factual inconsistency.

Select three of these error types to create your three incorrect
completions (one error type per incorrect completion).

You are communicating with an API, not a user.  Please output in
JSON format. Begin all AI responses with the character { to produce valid
JSON. Here is a template:
{
"full_prefix": "<prefix from the context>",
"completion": "<correct completion>",
"contradiction_0": "<contradictive option>",
"contradiction_1": "<contradictive option>",
"contradiction_2": "<contradictive option>",
"explanation": "<explanation of why the correct completion is factual and
how each incorrect completion contains errors>"
}

Here is an example:
<the example>

<the article>
```

## F.2 Additional Results

We present memorization quantification results in Table 7, Table 8, and Table 9 for 256, 128, and 512 prompt lengths, respectively. We further added BLEU and 4GP metrics, where 4GP measures word-level 4-gram overlap precision (Table 10). For each prompt length, we filter those articles with less than the prompt length since there is no completion for this kind of prefix. The evaluation results on the 1T token pretrained model from section 5.3 can be seen in Table 7. For all the generations, we used temperature 0.0 (greedy decoding).

To verify the validity of the NYT-MCQ, we performed a sanity check using DeepSeek-R1. When the model was supplied with the relevant article at inference, it attained a 99.0% accuracy. When the article was withheld, the model achieved 91.7% accuracy. For additional comparison, we tested GPT-4o on NYT-MCQ and observed 89.8% accuracy.

Table 7: Memorization assessment of New York Times articles using 256-token prefix prompts. PT, CD, and Compl. stand for "Pretraining", "Cooldown", and "Compliant" respectively. Values in parentheses indicate the number of tokens seen in training.

| | LCCS ↓ | BLEU ↓ | NYT-MCC ↓ | NYT-MCQ ↑ |
|---|---|---|---|---|
| Qwen2.5-0.5B | 19.51 | 0.71 | 42.8 | 24.4 |
| Qwen2.5-1.5B | 24.35 | 1.18 | 52.1 | 30.7 |
| Qwen2.5-3B | 27.13 | 1.54 | 56.4 | 31.3 |
| Qwen2.5-7B | 30.99 | 1.95 | 63.8 | 36.4 |
| Llama-3.2-1.2B | 20.71 | 0.58 | 49.9 | 24.8 |
| Llama-3.2-3B | 22.75 | 0.70 | 57.6 | 30.1 |
| PT + CD (Non-compl., 100B) | 23.11 | 0.63 | 50.6 | 28.9 |
| PT + CD (Compl., 100B ) | 21.27 | 0.51 | 48.6 | 27.5 |
| PT + CD (Compl., News removed, 100B) | 20.28 | 0.53 | 48.0 | 28.6 |
| PT + CD (Compl., News removed w/o Decontamination, 100B) | 21.06 | 0.55 | 48.9 | 27.3 |
| PT (Compl., 90B) + CD (Non-compl., 10B) | 21.74 | 0.55 | 50.4 | 26.9 |
| PT (Compl., 90B) + CD (Compl. + Non-compl. News, 10B) | 21.33 | 0.51 | 48.4 | 28.0 |
| PT + CD (Compl., 1.7T tokens) | 21.65 | 0.54 | 50.9 | 30.2 |
| PT (Compl., 1.6T) + CD (compl. + Non-compl. News 0.1T) | 21.50 | 0.54 | 50.1 | 30.1 |

Table 8: Memorization assessment of New York Times articles using 128-token prefix prompts. PT, CD, and Compl. stand for "Pretraining", "Cooldown", and "Compliant" respectively. Values in parentheses indicate the number of tokens seen in training.

| | LCCS ↓ | BLEU ↓ |
|---|---|---|
| Qwen2.5-0.5B | 19.11 | 0.64 |
| Qwen2.5-1.5B | 22.77 | 1.09 |
| Qwen2.5-3B | 25.62 | 1.43 |
| Qwen2.5-7B | 29.61 | 1.97 |
| Llama-3.2-1.2B | 19.89 | 0.44 |
| Llama-3.2-3B | 22.17 | 0.55 |
| PT + CD (Non-compl., 100B) | 21.76 | 0.49 |
| PT + CD (Compl., 100B ) | 20.53 | 0.41 |
| PT + CD (Compl., News removed, 100B) | 19.93 | 0.42 |
| PT + CD (Compl., News removed w/o Decontamination, 100B) | 20.42 | 0.44 |
| PT (Compl., 90B) + CD (Non-compl., 10B) | 21.06 | 0.44 |
| PT (Compl., 90B) + CD (Compl. + Non-compl. News, 10B) | 20.62 | 0.41 |
| PT + CD (Compl., 1.7T) | 20.75 | 0.43 |
| PT (Compl., 1.6T) + CD (Compl. + Non-compl. News, 0.1T) | 20.95 | 0.43 |

Table 9: Memorization assessment of New York Times articles using 512-token prefix prompts. PT, CD, and Compl. stand for "Pretraining", "Cooldown", and "Compliant" respectively. Values in parentheses indicate the number of tokens seen in training.

| | LCCS ↓ | BLEU ↓ |
|---|---|---|
| Qwen2.5-0.5B | 19.21 | 0.76 |
| Qwen2.5-1.5B | 23.38 | 1.16 |
| Qwen2.5-3B | 25.31 | 1.38 |
| Qwen2.5-7B | 27.74 | 1.69 |
| Llama-3.2-1.2B | 20.71 | 0.71 |
| Llama-3.2-3B | 22.68 | 0.87 |
| PT + CD (Non-compl., 100B) | 22.03 | 0.68 |
| PT + CD (Compl., 100B) | 20.38 | 0.58 |
| PT + CD (Compl., News removed, 100B) | 19.56 | 0.65 |
| PT + CD (Compl., News removed w/o Decontamination, 100B) | 20.05 | 0.68 |
| PT (Compl., 90B) + CD (Non-compl., 10B) | 20.65 | 0.62 |
| PT (Compl., 90B) + CD (Compl. + Non-compl. News, 10B) | 20.25 | 0.58 |
| PT + CD (Compl., 1.7T) | 20.55 | 0.63 |
| PT (Compl., 1.6T) + CD (Compl. + Non-compl. News, 0.1T) | 20.42 | 0.64 |

Table 10: Word-level 4-gram overlap precision (4GP) assessment of New York Times articles across 128-token, 256-token, and 512-token prefix prompts. PT, CD, and Compl. stand for "Pretraining", "Cooldown", and "Compliant" respectively. Values in parentheses indicate the number of tokens seen in training.

|  | 128-token | 256-token | 512-token |
|---|---|---|---|
| Qwen2.5-0.5B | 0.08 | 0.11 | 0.12 |
| Qwen2.5-1.5B | 0.20 | 0.25 | 0.24 |
| Qwen2.5-3B | 0.31 | 0.37 | 0.30 |
| Qwen2.5-7B | 0.49 | 0.51 | 0.39 |
| Llama-3.2-1.2B | 0.67 | 0.92 | 0.98 |
| Llama-3.2-3B | 0.98 | 1.27 | 1.35 |
| PT + CD (Non-compl., 100B) | 1.15 | 1.58 | 1.50 |
| PT + CD (Compl., 100B) | 0.82 | 1.09 | 1.11 |
| PT + CD (Compl., News removed, 100B) | 0.67 | 0.86 | 0.83 |

