# OpenReview forum: "Can Performant LLMs Be Ethical? Quantifying the Impact of Web Crawling Opt-Outs"
_colmweb.org/COLM/2025/Conference — COLM 2025_

### Official Review · Reviewer_mnmF · 2025-05-08

**Rating:** 7
**Confidence:** 4
**Ethics Flag:** 1

**Summary:**

This paper studies the "data compliance gap" (DCG), which refers to the performance gap between LLMs trained on data sets that comply with web crawling opt-outs vs those that do not. The authors consider two settings: 1) pretraining from scratch, 2) continual training from an existing compliant model. They find that, as of January 2025, compliance with web data opt-outs does not degrade general knowledge acquisition, but that for specialized domains, excluding major publishers (i.e., complying with opt-outs) leads to performance declines. The study gives some empirical insights into the effect of data compliance in 1.5B models, which may help future discussions on AI training practices and policy decisions.

**Questions To Authors:**

L263-264: What do the authors mean by "Variations within each column, excluding the underlined values, can be interpreted as noise or random fluctuations."? The underlying is also unclear to me.
-> Related to this remark: most accuracy differences seem (especially in Table 3, where the authors claim performance declines) quite small. Most experiments are computationally expensive; therefore, I understand there are no error bars, but I would not claim any significant differences here.

Have the authors tried "fake" experiments, pretending that even more pre-training data is non-compliant? Is there any indication of how much data needs to be removed before we see any performance degradation in the 1.5B LLMs?

Do the authors plan to share the code and filtered datasets?

**Reasons To Accept:**

The authors wrote a clear and technically sound paper. They acknowledge the limitations and conduct necessary robustness tests (such as memorization). The question of whether performant LLMs can be ethical is interesting and timely.

**Reasons To Reject:**

The authors acknowledge that a main limitation of this work is the relatively small scale of the LLMs, which may limit memorization and factual recall. Since most of their benchmarks are testing for this, it may not be as surprising that there are no "significant" (there are no error bars, see below in "questions to authors") DCG. Table 2, for example, indicates that compliant (and -news) data sets even lead to better performance on average? Additionally, the authors indicate that despite their removal techniques, there is still some non-compliant data present, which may further underestimate the effects.

---

> ### Author Response · Authors · 2025-06-02
>
> We thank the reviewer for recognizing our work as technically sound, timely and interesting.  Below, we address your questions and concerns. If any issues remain, please let us know, and we’ll be happy to provide further clarification.
>
> **Improved performance after News removal?** As shown in Table 2, performance improves after removing news content, which is unsurprising. Given that our training used only 100B tokens from the FineWeb-Edu dataset (a small fraction of the total 1T+ tokens available), removing the News domain increases the sampling rate for other domains. Please check our Figure 3, where Knowledge Article accounts for a higher percentage in the compliant corpus. This likely contributes to the observed performance gains, as it allows the model to focus more on other informative, high-quality data.
>
> **Significance of Cooldown experimental results:** Regarding your question on lines 263–264, we apologize for the lack of clarity. In Table 3, the underlined scores correspond to runs where task-specific, non-compliant data was added during cooldown, which is supposed to enhance performance on that specific task. The remaining entries illustrate expected variability across cooldown runs without such targeted augmentation. We would like to draw your attention to the bottom part of Table 3, where we cool down on a pretrained compliant checkpoint (1.6T tokens) with 100B tokens. Since a larger number of non-compliant domain-specific tokens are seen, we could observe the impact more clearly. For the medical domain experiments, we can observe a clear signal, taking the variation into account.
>
> In terms of fake experiments, yes, we conducted an additional experiment by deduplicating content to eliminate republished News articles. Please see our **global response** for more details.
>
> Yes, we will open-source the list of URLs and the compliant dataset upon acceptance.

---

> > ### Comment · Reviewer_mnmF · 2025-06-02
> >
> > Thank you for the clarification. I have upped my score.

---

### Official Review · Reviewer_Fr1t · 2025-05-13

**Rating:** 7
**Confidence:** 4
**Ethics Flag:** 1

**Summary:**

This paper studies the impact of web data provider opt-out on language model performance. The paper measures the performance gap between non-compliant models, which are trained on all available data, and compliant models, which do not train opt-out data. Many 1.5b-parameter Llama models are trained in two settings: pretraining from scratch, and introducing non-compliant data through continued pretraining of a compliant model. The paper finds that pre-training on compliant data does not lead to a significant decrease in model performance on general-purpose benchmarks.

**Questions To Authors:**

1. I think Figure 3 would be clearer if you showed the percentage decrease in each category---rather than the data distribution across topics---because I'd like to see how much of each domain is filtered out by compliance.
2. I find takeaway 3, that newspaper articles are not as important for downstream performance to be particularly surprising. I appreciate the additional investigation into news memorization in section 5.4.

Typos: Table 1's caption has a typo on 'percentage'. The last line of the intro should end in a question mark or be rephrased.

**Reasons To Accept:**

1. The paper shows clear results that training on compliant data does not lead to a large decrease in model performance. The experiments are well-designed, and the paper is clearly written.
2. The data compliance gap metric will likely be increasingly relevant as more data providers opt out of allowing their data to be trained on. The paper provides a framework for measuring the impact of data opt-out into the future.
3. The paper also shows how data opt-out impacts the distributions of topics and formats of pretraining data.

**Reasons To Reject:**

The main challenge of this paper is that the reported data compliance gaps are very small in all settings. This is an interesting finding in itself with ramifications for data curation. But it does raise the question of  how many of these gaps are due to data curation or due to other random variation in the training process (e.g. from data ordering, random seeds, etc). I think the paper presents a clear contribution as-is and mostly tailors its claims to its evidence (and most data ablation work does not train replicate models). However, it's not clear to me that Takeaway 4 is true without some estimate of variance (and variation in performance due to randomness is already alluded to in lines 263-264). I also recommend setting the y-axis in Figures 5 and 6 to start at 0 so as to not overstate differences between the models.

---

> ### Author Response · Authors · 2025-06-02
>
> We thank the reviewer for recognizing our contributions in introducing a framework to measure data compliance gaps and providing quantifiable insights into the impact of AI training opt-outs.  Below, we address your questions and concerns. If any issues remain, please let us know, and we’ll be happy to provide further clarification.
>
> **Small evaluation gaps and variations across runs**: We agree with your concerns over random variation in the training process, and we point out in Lines 263-264 as well this concern. As shown in Table 3 in our main text, the underlined scores correspond to runs where task-specific, non-compliant data was added during cooldown, which is supposed to enhance performance on that specific task. The remaining entries illustrate expected variability across cooldown runs without such targeted augmentation. We would like to draw your attention to the bottom part of Table 3, where we cool down on a pretrained compliant checkpoint (1.6T tokens) with 100B tokens. Since a larger number of non-compliant domain-specific tokens are seen, we could observe the impact more clearly. For the medical domain experiments, we can observe a clear signal, taking the variation into account.
>
> We will replace the y-axis in Figures 5 and 6 as suggested in the updated version.
>
> **Additional domain-specific statistics**: For additional details of Figure 3, we summarize the percentage drop as suggested in the two tables below. It is worth noting that while we notice some big drops in certain domains, they may only account for small portions of the whole corpus, such as Games and Politics. Science & Tech and Health are two major topic domains where a significant token loss due to compliance is observed. In terms of Format, the token drop is more concentratedly distributed than in terms of Topic.  News Article and Knowledge articles are the formats with the most significant token loss.
>
> In response to your comment regarding our takeaway3, we offer further experimental insights in our **global response**. Thanks for pointing out the typos. We will make sure these are addressed.
>
>
> Topic domain    | Percentage (of the domain) removed | Percentage (of the whole corpus) removed
> ------------------- |----------------------------------|----------------------------------------
> Science & Tech.      | 11.39  | 2.6483
> Health  | 9.65 | 1.8893
> History  | 8.05| 0.7921
> Education & Jobs  | 7.11 | 0.7639
> Politics | 10.45 |  0.4778
> Literature  | 8.94 | 0.3639
> Industrial | 8.02| 0.2438
> Home & Hobbies  | 6.01 | 0.2002
> Finance & Business   | 8.46 | 0.1894
> Art & Design  | 8.08 | 0.1832
> Transportation  | 10.84  | 0.1676
> Religion    | 3.58   | 0.1377
> Software Dev. | 6.52  | 0.1327
> Crime & Law   | 8.03 | 0.1184
> Food & Dining  | 8.46| 0.1114
> Software  | 6.92   | 0.1096
> Sports & Fitness | 7.36 | 0.0903
> Hardware  | 9.12 | 0.0829
> Entertainment | 9.44 | 0.0806
> Social Life  | 6.72| 0.0572
> Travel | 6.94 | 0.0534
> Games  | 11.69 | 0.0478
> Fashion & Beauty | 3.61 | 0.0089
> Adult | 0.81| 0.0001
>
>
>  Format domain  | Percentage (of the domain) removed | Percentage (of the whole corpus) removed
> --------------------|------------------------------------|------------------------------------------
>  News Article       | 24.09  | 2.4956
>  Knowledge Article  | 6.87 | 1.6224
>  Nonfiction Writing | 9.76 | 1.2853
>  Academic Writing   | 9.57| 0.7303
>  Tutorial | 5.95 | 0.5673
>  Product Page       | 6.53| 0.4566
>  Structured Data    | 17.39   | 0.3920
>  Q&A Forum   | 27.33  | 0.3518
>  Listicle  | 6.62 | 0.2115
>  Content Listing    | 8.51  | 0.1708
>  Personal Blog      | 1.68  | 0.1330
>  Truncated| 10.22 | 0.1236
>  News (Org.) | 2.49 | 0.0942
>  Audio Transcript   | 12.23  | 0.0940
>  Documentation      | 7.58 | 0.0804
>  FAQ  | 5.13| 0.0336
>  Comment Section    | 4.13 | 0.0233
>  Customer Support   | 2.88| 0.0223
>  About (Org.)   | 1.15  | 0.0208
>  Creative Writing   | 4.09| 0.0154
>  Spam / Ads  | 2.17  | 0.0122
>  User Review  | 4.09 | 0.0103
>  About (Pers.)  | 1.85| 0.0023
>  Legal Notices | 1.67 | 0.0020

---

> > ### Comment · Reviewer_Fr1t · 2025-06-07
> >
> > Thanks for your response!
> >
> > Thanks for the domain-specific statistics and the additional news experiments.
> >
> > I continue to think that small gaps combined with random variation is a confounder, and I'm glad that you mention this in lines 263-264. I agree with you that medical data in Table 3 in the 1.7T token setting shows the most performance difference, so I am okay with takeaway 4 as-is. I think lines 264-265 should clarify that medical tasks only show a clear signal in the 1.7T token setting and not the 100B token setting.

---

> > > ### Author Response · Authors · 2025-06-07
> > >
> > > Thank you for your suggestion! We agree that this should be stated more clearly.
> > >
> > > We will ensure that our updated manuscript clarifies that medical tasks only show a clear signal in the 1.7T token setting.

---

### Official Review · Reviewer_QefG · 2025-05-25

**Rating:** 10
**Confidence:** 4
**Ethics Flag:** 1

**Summary:**

This paper conducts a empirical study to quantify the tradeoff between data compliance and model performance by introducing the Data Compliance Gap metric. To study this, the authors create a compliant training corpus by filtering FineWeb-Edu based on robots.txt restrictions as of January 2025, then train 1.5B parameter models on both compliant and non-compliant versions. They evaluate performance differences using two approaches: training from scratch and continual pretraining with domain-specific data reintegration. The key finding is that compliant models achieve similar performance to the original model on general knowledge benchmarks, but show noticeable degradation in specialized domains, particularly medical knowledge.

**Questions To Authors:**

1. Just to double check: are both compliant and non-compliant models trained on the same amount of tokens? If you control for the same amount of tokens, does the compliant model see some repeating data since the compliant data pool is smaller than the non-compliant data?

2. As MMLU has different subject categories, I wonder how the model performance changes with respect to different categories. Maybe the compliant model would perform worse on STEM since the science-related data are opted out?

**Reasons To Accept:**

1. This is the first study to delve deep into how data compliance regulations affect LLM training, which is a very important question the community has been grappling with. It would be valuable if the authors could open-source their compliant datasets and the list of opted-out URLs.

2. The study is rigorous: the authors put a lot of efforts into curating the data properly and designing meaningful evaluations. The Data Compliance Gap concept they introduce seems like it could become a standard way for researchers to measure how much performance we lose when we respect opt-out requests.

**Reasons To Reject:**

1. The paper would benefit from more comprehensive domain-specific evaluations that better match the main categories of opted-out content: News & Media, Science & Technology, and Health & Medical Information. While the authors do include benchmark related to news, the paper would be stronger if more news domain evaluation is included given that news make up a significant portion of the filtered domains.

2. There could be some issues with equating robots.txt compliance with full data usage rights. Robots.txt only governs web crawling permissions, not training rights: a site allows crawling doesn't necessarily mean they consent to their content being used for AI training. The paper is measuring a upper bound of what could be considered compliant, but many sites that allow crawling might still prohibit AI training through their Terms of Service or other legal mechanisms. A deeper discussion of these broader compliance considerations would strengthen the paper's framing of what "compliance" actually means.

---

> ### Author Response · Authors · 2025-06-02
>
> Thank you for the positive and encouraging feedback. Below, we address your questions and concerns. If any issues remain, please let us know, and we’ll be happy to provide further clarification.
>
> **More News domain evaluation**: We provide an additional experimental result regarding News republication in our global response.
>
> **What "compliance" truly entails?** We acknowledge that robots.txt compliance represents only one dimension of data usage rights. Our approach goes beyond general crawling permissions by specifically evaluating robots.txt rules for *AI training-focused user agents* (including GPTBot, ClaudeBot, Google-Extended, and others as listed in Section B of our appendix), which provides a stronger signal of consent for AI training purposes than general crawling allowances. However, we recognise that this does not capture all legal mechanisms through which sites may restrict AI training, particularly Terms of Service accessible via “browsewrap agreements”. While such terms are legally more ambiguous in their enforceability and pose significant challenges for automated processing at scale due to their natural language format, we agree that a more comprehensive discussion of these broader compliance considerations would strengthen our paper's framing. We will expand our analysis to acknowledge these limitations and clarify that our robots.txt filtering represents a practical upper bound of demonstrable compliance rather than comprehensive legal clearance.
>
> Regarding the questions you raised, our responses are as follows:
>
> Q1: Yes, the compliant and non-compliant models are trained on the same amount of tokens, which is 100B here, sampled from over 1T tokens in FineWeb-Edu. Within our dataloader, the sampling state is tracked to ensure that each document is sampled at most once, with no repeated sampling.
>
> Q2: As suggested, we present the scores for MMLU subcategories as follows, where we follow the four supercategories from the original MMLU paper [1]. We did not observe lower scores in STEM, and we would like to highlight that the scores for STEM are around random (25%), which is expected given the scale of our experimental evaluation.
>
>   MMLU Task |  humanities | social sciences | stem | others
>  -------- |-------- |-------- |-------- |-------- |
> Non-compliant       | 28.9 |  36.0  | 27.1 | 37.7
> Compliant         | 28.4 | 36.6 | 27.5 | 37.3
> Exclude News | 28.7  |  35.4 | 27.9 | 36.8
>
> [1] Dan Hendrycks et al., Measuring Massive Multitask Language Understanding

---

### Author Response · Authors · 2025-06-02
**Global response**

We thank all reviewers for their insightful and encouraging feedback. A common concern appears to center on the experiments and analyses related to News articles. Below, we present additional results to offer further insights.

**Removal of Republished News Articles:** We conducted an additional experiment by deduplicating content to eliminate republished News articles (e.g., news reposted on personal blogs). As the News removal in our main text is only URL-based, there is a significant amount of republication on compliant websites (see Table 1 in our main text), even though these republished texts should not be crawled according to the original restrictions.  This deduplication led to an 18 % reduction in tokens from the compliant corpus, yet resulted in an even greater improvement in downstream performance. All models in this experiment were trained by sampling 100B tokens from their respective corpora. These results suggest that News articles may be less critical than previously assumed, provided that other high-quality content domains are present.

To assess memorization, we repeated the experiments from Section 5.4. We observed a further reduction in verbatim memorization, as measured by both LCCS and NYT-MCC (factual recall from a News excerpt).



  Task |  Arc-C | Arc-E | CSQA | OBQA | MMLU | PIQA | SIQA | HS  | LBD | WG | Avg
 -------- |-------- |-------- |-------- |-------- |-------- |-------- |-------- |-------- |-------- |-------- | --------
Non-compliant       | 34.1 |  70.0 | 20.8 | 27.4 | 32.0 | 71.5 | 40.4 | 42.0 | 34.7 | 52.2 | 42.5
Compliant         | 32.8 | 69.1 | 20.2 | 26.0 | 32.0 | 71.0 | 41.5 | 42.0 | 35.4 | 57.5 | 42.8
-News | 35.1 |  69.1 | 20.6 | 25.4 | 31.8 | 70.6 | 40.5 | 42.0 | 37.2 | 55.2 | 42.6
-News (decontaminated) |35.1 |70.1|19.8| 26.6|31.8|71.8|40.4|42.4|36.0|56.0|43.0


 Model                                         | LCCS ↓ | BLEU ↓  | NYT-MCC ↓ | NYT-MCQ ↑
 --------------------------------------------- | ------ | ------ | ----- | ---------
 Non-compliant                                 | 23.11  | 0.63    | 50.6      | 28.9
 Compliant                                     | 21.27  | 0.51  | 48.6      | 27.5
-News          | 21.06  | 0.55    | 48.9      | 27.3
-News (decontaminated)     | 20.14  | 0.52   | 46.8      | 28.5

---

### Decision · Program_Chairs · 2025-07-08

**Decision:**

Accept

**Comment:**

This paper conducts a empirical study to quantify the tradeoff between data compliance and model performance by introducing the Data Compliance Gap metric. To study this, the authors create a compliant training corpus by filtering FineWeb-Edu based on robots.txt restrictions as of January 2025, then train 1.5B parameter models on both compliant and non-compliant versions. They evaluate performance differences using two approaches: training from scratch and continual pretraining with domain-specific data reintegration. The key finding is that compliant models achieve similar performance to the original model on general knowledge benchmarks, but show noticeable degradation in specialized domains, particularly medical knowledge.

Reasons To Accept:
* This is the first study to delve deep into how data compliance regulations affect LLM training, which is a very important question.
* The study is rigorous: the authors put a lot of efforts into curating the data properly and designing meaningful evaluations.
* The paper shows clear results that training on compliant data does not lead to a large decrease in model performance.
* The data compliance gap metric will likely be increasingly relevant as more data providers opt out of allowing their data to be trained on. The paper provides a framework for measuring the impact of data opt-out into the future.
* The paper also shows how data opt-out impacts the distributions of topics and formats of pretraining data.

Weaknesses / Future Work: (no reason to reject)
* There could be some issues with equating robots.txt compliance with full data usage rights. Robots.txt only governs web crawling permissions, not training rights: a site allows crawling doesn't necessarily mean they consent to their content being used for AI training.
* It would be interesting to consider a SOTA data mix that goes beyond just web text, an a broader notion of copyright, but clearly out of scope for now.
* The authors acknowledge that a main limitation of this work is the relatively small scale of the LLMs, which may limit memorization and factual recall.